# The Preparation of High-Performance and Stable MXene Nanofiltration Membranes with MXene Embedded in the Organic Phase

**DOI:** 10.3390/membranes12010002

**Published:** 2021-12-21

**Authors:** Qiang Xue, Kaisong Zhang

**Affiliations:** 1Key Laboratory of Urban Pollutant Conversion, Institute of Urban Environment, Chinese Academy of Sciences, Xiamen 361021, China; qxue@iue.ac.cn; 2University of Chinese Academy of Sciences, Beijing 100049, China

**Keywords:** MXene, nanofiltration, anti-swelling, mono/divalent ion separation, high saline removal

## Abstract

Nanomaterials embedded in nanofiltration membranes have become a promising modification technology to improve separation performance. As a novel representation of two-dimensional (2D) nanomaterials, MXene has nice features with a strong negative charge and excellent hydrophilicity. Our previous research showed that MXene nanosheets were added in the aqueous phase, which enhanced the permeselectivity of the membrane and achieved persistent desalination performance. Embedding the nanomaterials into the polyamide layer through the organic phase can locate the nanomaterials on the upper surface of the polyamide layer, and also prevent the water layer around the hydrophilic nanomaterials from hindering the interfacial polymerization reaction. We supposed that if MXene nanosheets were added in the organic phase, MXene nanosheets would have more negative contact sites on the membrane surface and the crosslinking degree would increase. In this study, MXene were dispersed in the organic phase with the help of ultrasound, then MXene nanocomposite nanofiltration membranes were achieved. The prepared MXene membranes obtained enhanced negative charge and lower effective pore size. In the 28-day persistent desalination test, the Na_2_SO_4_ rejection of MXene membrane could reach 98.6%, which showed higher rejection compared with MXene embedded in aqueous phase. The results of a long-time water immersion test showed that MXene membrane could still maintain a high salt rejection after being soaked in water for up to 105 days, which indicated MXene on the membrane surface was stable. Besides MXene membrane showed high rejection for high-concentration brine and good mono/divalent salt separation performance in mono/divalent mixed salt solutions. As a part of the study of MXene in nanofiltration membranes, we hoped this research could provide a theoretical guidance for future research in screening different addition methods and different properties.

## 1. Introduction

Desalination of seawater and brackish water has become one of the most sustainable methods to alleviate water shortages, and brackish water contains more divalent salts such as SO_4_^2−^ and Mg^2+^ compared with seawater [1]. As a typical representation of pressure-driven membranes, nanofiltration (NF) has been widely used in wastewater treatment, water softening, food processing and chemical processes [2]. NF membranes have excellent separation performance in the process of salt separation and recovery, due to the physical and chemical properties between NF membranes and ions. Moreover, NF membranes have a nice separation performance for monovalent and divalent ions and it has attracted increasing attention in mono/divalent ions separation [3].

By contrast with the dissolution diffusion separation mechanism of reverse osmosis membranes, the selectivity and recovery of salt ions in NF membranes depend on the hydration size and charge density of ions. Therefore, the NF membrane desalination is based on a comprehensive process of size sieving and charge effects. The working principle of the membrane charge effect is to separate divalent ions from mono/divalent ions with the help of the Donnan effect. The membrane will reject ions with higher charges, and lower charges ions will be transferred to the permeate side to maintain neutral electricity on both sides of the membrane [4]. Therefore, charge modification of the membrane surface can effectively improve the mono/divalent ions separation performance of the membrane. Therefore, current membranes separation performance relies on how to effectively enhance membrane surface charge effect. Although NF membranes have good separation performance, the membranes are limited by the “trade off” phenomenon and the serious operating environments (such as high-concentration brine) [4,5]. When exposed to high-concentration brine, the rejection of NF membranes is not as good as in conventional solutions concentration. At high-concentration brine, the membrane charge caused by the ionization from membrane functional groups is greatly affected, and the electrostatic repulsion effect between the charged solute and the original membrane is less effective [4]. This also indicated a stronger membrane charge force needed for high-concentration brine.

At present, the most advanced nanofiltration membranes are prepared through interfacial polymerization and a polyamide (PA) layer is deposited on a porous support layer [6]. The PA layer controls the physicochemical properties and separation performance of the NF membrane. By regulating the PA layer, a series of multifunctional NF membranes becomes possible. Embedding nanomaterials in thin-film composite membranes to prepare thin film nanocomposite (TFN) membranes is an effective modification technology to enhance performance [7]. Among many nanomaterials, two-dimensional (2D) nanomaterials have caused widespread attentions due to their atomic-level thickness, precise nano-layer spacing, strong mechanical strength, excellent chemical stability and adjustable surface sites [8]. MXene is a new type of 2D transition metal carbon/nitride discovered 10 years ago. Owing to its strong negative charge, excellent hydrophilicity and adjustable properties, it has been widely used in membrane modification to improve desalination performance [9,10,11,12,13]. MXene can become an excellent modifier for membrane performance.

The additional methods of nanomaterials included insertion in the aqueous phase or the organic phase for interfacial polymerization. The physicochemical characteristics and performance might differ from different additional methods when employing the same nanomaterials, such as TiO_2_ [14,15]. Our previous research showed that MXene nanosheets were added in the aqueous phase, which improved the permeselectivity of the membrane and achieved persistent desalination performance. MXene was a hydrophilic nanomaterial and the embedding of MXene in the aqueous phase would cause a water layer to form on the MXene surface, which hindered the interfacial polymerization reaction and formed nanovoids [16]. Although the flux could be improved, the rejection was not much higher for ions passing easily through nanovoids [17]. Moreover, the MXene nanosheets were more covered by the PA layer when added in the aqueous phase, and the improvement of rejection performance was limited. We hypothesized that if MXene nanosheets were added in the organic phase, more MXene nanosheets would appear on the membrane surface and the negative charge of the membrane would be directly enhanced. Due to the lack of the water layer, MXene embedded in the organic phase would have higher crosslinking degree.

In this research, we assumed that hydrophilic and negative MXene nanomaterials were embedded in the organic phase for enhanced negative charge and lower effective pore size compared with previous research. So far, the different performance of the same nanomaterials in the aqueous phase and organic phase of interfacial polymerization has not been explored. MXene nanomaterials were added in the organic phase, on the one hand for maximizing the charge effect of MXene and the charged solute and on the other hand for promoting the reaction degree of interfacial polymerization. Combined with previous research about MXene embedded in aqueous phase, we hoped that the same nanoparticles with two different methods would provide a theoretical guidance for future research in screening different addition methods and different properties.

## 2. Experimental Section

### 2.1. Materials

Previously reported PSf ultrafiltration (UF) membranes were adopted for the base membrane [17]. The pure water flux of the UF membrane is 331 Lm^−2^h^−1^bar^−1^, and the BSA rejection is 98.8%. Hydrochloric acid (HCl, 36–38%, Sino-pharm, Shanghai, China) and lithium fluoride (LiF, 99%, Macklin, Shanghai, China) were used for in-situ synthesis of hydrofluoric acid (HF), and etching titanium aluminum carbide (Ti_3_AlC_2_, 200 mesh, 98%, Macklin, Shanghai, China) to obtain Ti_3_C_2_T_x_. The aqueous phase monomer was piperazine (PIP, 99%, Aladdin, Shanghai, China), the acidic adsorbent was triethylamine (TEA, 99%, Aladdin, Shanghai, China) and (±)-camphor-10-sulfonic acid (CSA, 99%, Aladdin, Shanghai, China) for adjusting the pH, sodium dodecyl sulfate (SDS, 99%, Macklin, Shanghai, China) was used as a dispersant in the aqueous phase. The organic phase monomer was 1,3,5-benzenetrimethyl sodium chloride (TMC, 98%, Aladdin, Shanghai, China), and n-hexane (97%, Aladdin, Shanghai, China) was the TMC solvent. Use sodium sulfate (Na_2_SO_4_, 99%, Macklin, Shanghai, China), magnesium sulfate (MgSO_4_, 99.9%, Macklin, Shanghai, China), magnesium chloride (MgCl_2_, 99%, Macklin, Shanghai, China), sodium chloride (NaCl, 99.5%, Macklin, Shanghai, China) for separation performance test, and potassium chloride (KCl, 99.99%, Macklin, Shanghai, China) was tested for zeta potential. The molecular weight cut-off (MWCO) of the NF membrane was tested with 200, 300, 400, 600 and 1000 Da polyethylene glycol (PEG, Sino-Pharm, Shanghai, China). The water adopted in the experiment was deionized (DI) water.

### 2.2. Preparation of MXene

The preparation method of Ti_3_C_2_T_x_ nanosheets was based on the previously reported synthesis method [18]. Mixed 2 g LiF and 20 mL 9 molL^−1^ HCl in a 50 mL plastic bottle and stirred for 10 min to prepare an HF etchant solution. In order to let Ti_3_AlC_2_ fully peel off, this reaction prolonged the synthesis time. Then we slowly added 2 g Ti_3_AlC_2_ to the etching solution and stirred at 35 °C for 72 h to make the reaction complete. The acidic product was washed repeatedly with deionized water by centrifugation (5 min/cycle, 3500 rpm), and the pH was 6 after washing five times. The prepared Ti_3_C_2_T_x_ nanosheets were freeze-dried and used for subsequent research.

### 2.3. Preparation of MXene Membrane

The PSf UF membrane used for interfacial polymerization was immersed in DI water for 12 h in advance to remove the residual sodium bisulfite. The various additives in the aqueous phase were: 1.50 wt% CSA, 1.50 wt% TEA, 0.05 wt% SDS, and 1.50 wt% PIP. The organic phase was n-hexane solution containing 0.45 wt/v% TMC, and different MXene addition amounts (0 ppm, 100 ppm, 140 ppm, 160 ppm, 180 ppm, 200 ppm, 220 ppm, 340 ppm) were added in the organic phase. Before interfacial polymerization, the organic phase solution containing MXene was ultrasonically dispersed for 12 h in advance. The specific preparation process of the MXene membrane: first deposit the aqueous solution on the PSf UF membrane for 80 s, then remove the remaining aqueous solution and air dry. Then it was soaked in the TMC organic phase for 25 s, then placed in an oven at 80 °C for 1 min, and finally stored in DI water for further use. The synthesis process of MXene nanocomposite membrane was shown in Figure 1.

### 2.4. Characterization of MXene

Scanning electron microscope (FE-SEM, S-4800, HITACHI, Tokyo, Japan) was used to observe the morphology of Ti_3_AlC_2_ and Ti_3_C_2_T_x_, and transmission electron microscope (TEM, Talos-S, FEI, MA, USA) was used to observe the layered structure and atomic force microscope (AFM, Dimension Icon, Bruker, WI, USA) was used to observe the thickness of Ti_3_C_2_T_x_. X-ray diffractometer (XRD, X’Pert Pro, PANalytical, Lelyweg, The Netherlands) was used to reveal the crystal structure and d-spacing of Ti_3_C_2_T_x_. Energy-dispersive X-ray spectroscopy (EDXS) and X-ray photoelectron spectroscopy (XPS, Axis Supra, Kratos, Manchester, UK) were used to analyze the elemental composition of Ti_3_C_2_T_x_ nanosheets. Fourier infrared spectroscopy (FTIR, iS10, Thermo, Waltham, MA, USA) was used to observe the functional group structure of Ti_3_C_2_T_x_.

### 2.5. Characterization of MXene Membrane

All the characterized membranes were dried in a freeze dryer (FD-1A-50, BIOCOOL, Beijing, China) for 48 h before characterization. SEM was used to observe the surface and cross-section of the films, XRD and EDXS were used to observe the crystal structure of the film surface and analyze the element composition of the membrane surface. The surface roughness and morphology of the membranes were obtained by AFM observation. The hydrophilicity of the membrane surface was evaluated by a contact angle analyzer (DSA100, KRUSS, Hamburg, Germany), and the surface charge was tested by a zeta potential instrument (SurPASS 3, Anton Paar, Glaz, Austria). Fourier infrared spectroscopy (FTIR, iS10, Thermo, Waltham, MA, USA) was used to observe the functional group structure of the nanofiltration membrane. X-ray photoelectron spectroscopy (XPS, Axis Supra, Kratos, Manchester, UK) was used to determine the elemental composition of the membranes.

### 2.6. Separation Performance of MXene Membrane

A dead-end membrane filter tank (Amicon Stirred Cell, Millipore, Burlington, MA, USA) was used to test the separation performance of the NF membranes, which was more convenient than a complex cross-flow system. Under 4 bars operating pressure, the experimental data obtained by analyzing the balance would be automatically recorded in the computer. For a single salt solution, 2000 ppm Na_2_SO_4_, MgSO_4_, MgCl_2_, and NaCl solutions were used as feed solutions. Before the test, each membrane was pre-compressed at a pressure of 4 bars for 30 min. For the mixed solutions of Na_2_SO_4_/NaCl and MgSO_4_/NaCl, the single salt concentration in two mixed salt systems was 1000 ppm. For the desalination test of high-concentration brine, the testing pressure was adjusted to 6 bars due to the influence of greater concentration polarization, and the other test parameters remained unchanged. The permeation flux was calculated by the Formula (1):(1)$$J_{w}=\frac{Q}{At}$$
where $J_{w}$ (Lm^−2^h^−1^) was the permeation flux, Q was the total permeated water volume during the period (L) in t (h), A was the membrane test area (m^2^).

During the desalination test, the membrane was pre-compressed for 30 min and then the permeate was collected, and the conductivity of the permeate was measured using a conductivity meter (DDS-307A, INESA, Shanghai, China). The separation performance was calculated by Formula (2):(2)$$R\left(\%\right)=\left(1-\frac{C_{p}}{C_{f}}\right)\times 100\%$$
where $C_{p}$ and $C_{f}$ were the salt concentration of the permeate and the feed solution, respectively.

PEG solutions with different molecular weights (200, 300, 400, 600, and 1000 Da) with a concentration of 1000 ppm were used to test the MWCO of the NF membranes. The rejection rate calculation method was the same as that of the salt solution test. When the rejection reached 90%, the corresponding molecular weight was the membrane MWCO. Using 2000 ppm Na_2_SO_4_ salt solution, the MXene membrane was carried out for 105-day water immersion desalination test and 28-day persistent desalination test under 4 bars pressure. In the persistent desalination test, JJ0 was used to evaluate the permeation flux change, J was the permeation flux per time, and $J_{0}$ was the initial permeation flux (take the average of the previous three times as $J_{0}$). We collected 134 data points during the entire persistent desalination process. 

## 3. Results and Discussions

### 3.1. Physicochemical Characterization of MXene 

As shown in Figure 1a, the morphology of Ti_3_AlC_2_ before etching was a solid block. The Ti_3_C_2_T_x_ obtained showed relatively small and thin fragments on porous alumina support. The lateral size of Ti_3_C_2_T_x_ under SEM was within 1 μm (Figure 1b). Figure 1c showed the morphology of Ti_3_C_2_T_x_ under TEM. It could be observed that Ti_3_C_2_T_x_ nanosheets had a 4-layer structure. The thickness and morphology analysis of Ti_3_C_2_T_x_ nanosheets in AFM were shown in Figure 1d,e. The measured thickness of Ti_3_C_2_T_x_ nanosheets was 3.5 nm. Considering the reported single layer thickness of Ti_3_C_2_T_x_ was 0.84 nm, the prepared Ti_3_C_2_T_x_ nanosheets had a 4-layer structure [12]. The AFM results were consistent with the TEM results, which further proved that the prepared Ti_3_C_2_T_x_ nanosheets had a thinner laminated structure.

As shown in Figure 2a, the characteristic peak (002) of generated Ti_3_C_2_T_x_ after etching moved from 9.6° to 7.7°, and the (104) peak basically disappeared, indicating that Ti_3_AlC_2_ was well been etched. Combining the Bragg equation with the previously reported monolayer Ti_3_C_2_T_x_ thickness, the interlayer spacing between Ti_3_C_2_T_x_ nanosheets was 0.306 nm [12]. Energy-dispersive X-ray spectroscopy was used for Ti_3_C_2_T_x_ elemental composition analysis and the Al element in Ti_3_AlC_2_ was not detected as shown in Figure 2b. It further showed that Ti_3_AlC_2_ was well peeled off. Figure 2c showed the XPS full spectrum analysis of Ti_3_C_2_T_x_. In addition to Ti and C elements, Ti_3_C_2_T_x_ also obtained O, F and Cl after HF etching. Figure 2d showed that the strongest peak on the FTIR map of Ti_3_C_2_T_x_ was at 3420 cm^−1^, which corresponded to a –OH functional group. The peak at 1630 cm^−1^ was a C=O functional group, and 1110 cm^−1^ was a C–F functional group. It presented that the prepared Ti_3_C_2_T_x_ had abundant oxygen-containing and fluorine-containing functional groups, which made Ti_3_C_2_T_x_ have a higher negative charge [19]. At the same time, the abundant functional groups were conducive to the formation of hydrogen bonds between the nanosheets and hydrated ions, and hydrogen bonds had potential advantages in improving the desalination performance of the membrane [20].

### 3.2. Characterization of MXene Membrane

The XRD results of TFC membrane and membrane with 180 ppm MXene were shown in Figure 3a. Compared with TFC membrane, the MXene membrane had a characteristic peak of MXene (002) with a 2θ of 6.3°, which indicated that the membrane surface was loaded with a certain amount of Ti_3_C_2_T_x_. MXene nanosheets loaded on the membrane surface ensured direct contact with the charged solute for higher rejection. By subtracting the thickness of the monolayer nanosheet from the d-spacing value calculated by the Bragg equation, it was expected that the layer spacing between the two-dimensional nanosheets on the membrane was about 0.55 nm. The above results were basically consistent with the previous report [21]. This sub-nanometer free layer spacing suggested that MXene membranes could be used to separate ions through ion transport mechanisms [21]. The EDXS analysis of membrane surface with 180 ppm MXene was shown in Figure 3b,c, and Ti and F elements from Ti_3_C_2_T_x_ could be observed. Compared with the EDXS analysis of the previous MXene membranes prepared in aqueous phase, the elemental content of Ti and F increased significantly, which meant that more MXene nanosheets covered the membrane surface [17]. Based on the above characterization results, MXene nanosheets were successfully embedded in the PA layer and an effective nanochannel was formed for rejecting ions.

The membrane surface and cross-sectional morphology are shown in Figure 4a, due to the fast reaction between PIP and TMC, the surface of TFC membrane presented a representative bubble structure, which was coherent with the previous report [22]. In Figure 4b, the bubble size of the membrane surface with 180 ppm MXene was smaller and denser than that of the TFC membrane, indicating that the MXene membrane had a higher degree of surface cross-linking than the TFC membrane. This might be due to the reaction of the abundant hydroxyl functional groups of MXene and unreacted acid chloride, reducing the carboxylic acid formed by the final hydrolysis of unreacted acid chloride and enhancing the crosslinking degree of interfacial polymerization [23]. On the surface of the MXene membrane, it could be seen that MXene nanosheets were evenly distributed on the surface of the membrane, and some areas of the nanosheets were covered by polymer, which could enhance the stability and anti-swelling properties of 2D MXene membranes. More MXene nanosheets were covered on the membrane to facilitate direct contact with target solute.

The cross-sectional morphology of TFC membrane and membrane with 180 ppm MXene was shown in Figure 4c,d. The cross-section of the MXene membrane was thinner and rougher than the TFC membrane. The surface roughness and morphology of membranes were observed through the 2D and 3D modes of AFM. As shown in Figure 5, MXene nanosheets were observed on the membrane surface, and the inorganic nanomaterials exhibited a white structure under AFM [24]. This also confirmed that the addition of MXene nanosheets in the organic phase made MXene nanosheets appear on the membrane surface. The surface roughness of MXene membrane increased with the increase of MXene nanosheets addition and was rougher than the original TFC membrane.

Due to the addition of MXene nanomaterials with good hydrophilicity, the membrane surface hydrophilicity has been greatly improved. As the adding amount of MXene increased, the water contact angle of the membrane decreased from 45.2° to 34.5° in Figure 6a, which was coherent with previous reports on Mxene [17]. Besides, MXene nanomaterials had a higher negative surface charge. After MXene was added in the interfacial polymerization, the negative charge on the membrane surface could be significantly improved [11,25]. Figure 6b shows that the zeta potential of the membrane with 180 ppm MXene was obviously lower than that of TFC membrane, which was conducive to improving the anion rejection. The results showed that more negatively charged MXene covered the membrane surface, which could further reduce the zeta potential on the membrane surface. The FTIR analysis of TFC membrane and membrane with 180 ppm MXene was shown in Figure 6c. The stretching vibrations of the –OH group at 3365 cm^−1^, 3484 cm^−1^, and 3598 cm^−1^ were slightly increased. The peaks 1426 cm^−1^ and 1540 cm^−1^ corresponding to the amide bond of membrane with 180 ppm MXene were weaker than the TFC membrane. This might be due to the abundant –OH brought by the nanomaterials, which made the characteristic peaks of polyamide exposed on the surface reduced [12]. The XPS full-spectrum scan result of the diaphragm was shown in Figure 6d. Compared with the TFC membrane, the content of O and N in MXene membrane was significantly enhanced. The increase of O element was attributed to the abundant oxygen-containing functional groups in MXene nanomaterials, and the increase of N element content were attributed to the increase of polyamide formed by interfacial polymerization, reflecting the increase of the crosslinking degree of the membrane surface.

### 3.3. Separation Performance of MXene Membrane

After the addition of MXene nanomaterials, the Na_2_SO_4_ rejection of MXene membrane has been significantly improved (Figure 7a). This might be owing to the increase in the cross-linking degree and negative charge in the membrane surface with the help of negative MXene, which increased the rejection of divalent anions [26]. A similar mechanism has also been reported in MoS_2_ and GO [23,27]. As shown in Figure 7b–d, the membrane with 180 ppm MXene had a certain increase in the rejection of MgSO_4_, MgCl_2_ and NaCl compared with TFC membrane. This might be attributed to the increase in the cross-linking degree of interfacial polymerization, making the effective pore size of the membrane smaller and increasing the ions’ rejection. In addition, the interlayer channels in the MXene membrane might have had hydrogen bonding sites, which had a hydrogen bonding effect with hydrated ions for hindering hydrated ions rapidly pass through the membrane [20]. 

The pure water flux of the membrane was presented in Figure 8a. Thanks to the increase in the hydrophilicity and roughness of the membrane surface, the water contact force and the effective filtration area were enhanced. The pure water flux of MXene membrane achieved a certain improvement. At the same time, the membrane with 180 ppm MXene could also maintain an equivalent pure water flux compared to TFC membrane and the highest rejection. In order to determine the effective pore size of the MXene membrane, the molecular weight cut-off test was performed on TFC membrane and the membrane with 180 ppm MXene, and the test results are shown in Figure 8b. The molecular weight cut-off of the membrane with 180 ppm MXene was 226 Da, which was close to the lowest reported the molecular weight of the NF membrane [28]. The molecular weight cut-off of TFC membrane was 289 Da, and the effective pore size of the membrane with 180 ppm MXene was reduced from 0.39 nm of TFC membrane to 0.34 nm. This also verified that the cross-linking degree of the MXene membrane increased and the effective pore size became smaller. The effective aperture of the TFN membrane with MXene in the organic phase was lower than the 0.38 nm aperture of the membrane with MXene in the aqueous phase. Embedding MXene nanosheets in the organic phase was more conducive to promoting the cross-linking of interfacial polymerization.

The filtration test results of different concentrations of Na_2_SO_4_ were shown in Figure 8c. 1000 ppm and 2000 ppm were the most commonly used concentrations of the NF membrane desalination test, but the salt concentration in the actual aqueous environment was much higher than these. Since ions in a high concentration solution easily penetrated the membrane surface, it often resulted in a lower salt rejection. The membrane with 180 ppm MXene could maintain a high rejection of more than 98% from 1000 ppm to 8000 ppm, and the rejection could reach 96.7% even at 10000 ppm salt solution. The selectivity of MXene membrane maintained a high level when the salt concentration increased, which showed its huge application advantage in the field of high-concentration saline. The rejection of the membrane with 180 ppm MXene in mono/divalent mixed salt system was shown in Figure 8d. In the Na_2_SO_4_/NaCl mixed salt system with a concentration of 1000 ppm, respectively, the membrane with 180 ppm MXene maintained a high rejection of 96.3% for Na_2_SO_4_, while the rejection rate for NaCl is only 8.2%, and the Cl/SO_4_^2−^ selectivity coefficient was 24.8. The lower rejection of NaCl might be due to the fact that monovalent ions could more easily enter the nanochannel with water molecules and more Na^+^ accumulated in the ion-selective channel, which reduced the membrane’s Donnan effect on monovalent ions, resulting in rapid permeation of monovalent ions [29]. In the MgSO_4_/NaCl mixed salt system, the membrane with 180 ppm MXene could also maintain a high selectivity coefficient of mono/divalent salts and the Cl/SO_4_^2−^ separation coefficient was 44.2. The high rejection of divalent ions and the low rejection of monovalent ions might be owing to the adsorption and aggregation of monovalent ions in the ion selective channels of MXene nanosheets, which reduced the size of ion channels and made it difficult for divalent ions to pass through [30]. The separation results of monovalent and divalent salts showed that MXene membrane had excellent monovalent and divalent separation capabilities.

### 3.4. Long-Time Water Immersion Test of MXene Membrane

In order to evaluate the anti-swelling properties of MXene membrane, the prepared membrane with 180 ppm MXene was immersed in water for 105 days, and the permeate flux and rejection of the membrane at different stages was tested. As shown in Figure 9, the Na_2_SO_4_ rejection of membrane with 180 ppm MXene remained stable in each stage and could reach 98%. After immersed in water for 105 days, the Na_2_SO_4_ rejection could reach 97.3%, which was very near to the initial rejection. The Permeate flux remained a relatively stable level throughout the test. The long-time water immersion test of MXene membrane showed that the MXene membrane overcame the defect that the MXene membrane prepared under traditional auxiliary filtration was easy to swell and disperse. The experimental results showed that the 2D film prepared by embedding MXene nanosheets into the selective layer through interfacial polymerization had good anti-swelling properties and stability.

### 3.5. Stability Test of MXene Membrane

Under the pressure of 4 bars, 2000 ppm Na_2_SO_4_ was adopted to perform a 28-day persistent desalination test on the membrane with 180 ppm MXene. As shown in Figure 10, during the entire desalination process, the Na_2_SO_4_ rejection remained quite stable and at a high level, and the Na_2_SO_4_ rejection could reach 98.6%. The permeation flux of the membrane also maintained a stable level. The average J/J_0_ in the whole process could reach 0.987, which has little change compared with the initial permeation flux. Figure 11a,b showed the surface morphology of MXene membrane after 28-day filtration and the initial MXene membrane surface. The comparison represented that the filtered MXene membrane surface was not covered by many contaminants, and some MXene nanosheets could still be observed. Due to the long-term pressure environment, the filtered MXene membrane surface did not have as many bubble structures as the pristine MXene membrane, besides the membrane surface appeared more compact. The desalination performance comparison of the TFN membrane prepared by other 2D nanosheets was shown in Table 1. The MXene membrane in this study could maintain the highest Na_2_SO_4_ rejection under the conditions of low operating pressure, high feed liquid concentration and long running time. The Na_2_SO_4_ rejection was close to 99% of the ideal rejection for Na_2_SO_4_ desalination.

### 3.6. Performance Comparison of MXene Membranes with Two Different Additions Way

The performance comparison of MXene membranes with two different additions way was shown in Table 2. The MWCO of MXene membrane with MXene in organic phase was lower than MXene in aqueous phase, which indicated the effective aperture was also lower and the surface of the MXene membrane with MXene in organic phase was denser. MXene in the organic phase promoted the reaction degree of interfacial polymerization. At the same time, the Na_2_SO_4_ rejection of MXene membrane with MXene in organic phase was higher than that in aqueous phase. This might be attributed to the lower effective aperture and enhanced negative charge in membrane surface. However, the flux of MXene membrane in the aqueous phase was higher than that in the organic phase. In the aqueous phase, the surface of the hydrophilic nanoparticles would form a water layer so that the interfacial polymerization would form a nano-porous intermediate PA layer [16]. A loose PA structure made a contribution to increasing flux. The long-duration time of MXene membrane with MXene in the organic phase was shorter than that in aqueous phase. That’s because MXene was more located on the membrane surface and easier to lose than in the aqueous phase.

## 4. Conclusions

The MXene nanomaterial with strong negative charge was embedded in the organic phase, then the prepared MXene nanocomposite nanofiltration membrane showed enhanced surface charge, and the effective aperture was further reduced. In the 28-day persistent desalination test, the Na_2_SO_4_ rejection of MXene membrane could reach 98.6%, which showed higher rejection compared with MXene embedded in aqueous phase. The results of long-time water immersion test showed that MXene membrane could still maintain a high salt rejection after being soaked in water for up to 105 days, which indicated MXene on the membrane surface was stable although more MXene covered in membrane surface. Moreover, the MXene membrane showed high rejection for high-concentration brine and good mono/divalent salt separation performance in mono/divalent mixed salt solutions. This research also discussed different performances of MXene membranes with two different additions. We hoped that the same nanoparticles employed with two different methods would provide theoretical guidance for future research in screening different methods and different properties.

## Data Availability

Not applicable.

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
