# Peer review of "The Preparation of High-Performance and Stable MXene Nanofiltration Membranes with MXene Embedded in the Organic Phase"

_membranes, 2021, doi:10.3390/membranes12010002_

Round 1

Reviewer 1 Report

This manuscript would be of some interest to the readers. However, before accepted, the authors should fix the following issues:

  1. The Abstract of this manuscript is too long, and the authors need to make it shorter.
  2. The discussion about FIIR in Figure 6c is not convincing enough. If the authors just compared stretching vibrations of the -OH group at 3365cm-1 , 3484cm-1 , and 3598cm-1, please make it clearly.
  3. The x-axis (Figure 6a, Figure 7, Figure 8a) should revise to the contents of MXene rather than the sample name. This will make the results be clearly presented. The similar revion should be done on Figure 9.

Author Response

Reviewer 1’s Comments

This manuscript would be of some interest to the readers. However, before accepted, the authors should fix the following issues:

  1. The Abstract of this manuscript is too long, and the authors need to make it shorter.

Response: Thanks for this kind comment. We follow your comments and simplify the abstract. The revised version is visible in the revised manuscript. Compared with the previous abstract, the new abstract has reduced 104 words.

  1. The discussion about FTIR in Figure 6c is not convincing enough. If the authors just compared stretching vibrations of the -OH group at 3365cm-1, 3484cm-1, and 3598cm-1, please make it clearly.

Response: The previous discussion lacks sufficient consideration and it has been corrected. Revision shown as: The FTIR analysis of M0 and M3 was shown in Figure 6(c). The stretching vibrations of the -OH group at 3365 cm-1, 3484 cm-1, and 3598 cm-1 were slightly increased. The peaks 1426 cm-1 and 1540 cm-1 corresponding to the amide bond of M3 were weaker than M0. This might be due to the abundant -OH brought by the nanomaterials, which made the characteristic peaks of polyamide exposed on the surface reduced [12].

References added in revised version:

[12] X. Wang, Q. Li, J. Zhang, H. Huang, S. Wu, Y. Yang, Novel thin-film reverse osmosis membrane with MXene Ti3C2Tx embedded in polyamide to enhance the water flux, anti-fouling and chlorine resistance for water desalination, Journal of Membrane Science, 603 (2020) 118036.https://doi.org/10.1016/j.memsci.2020.118036

  1. The x-axis (Figure 6a, Figure 7, Figure 8a) should revise to the contents of MXene rather than the sample name. This will make the results be clearly presented. The similar revsion should be done on Figure 9.

Response: We had made a correction following your suggestion. The revised version is visible in the revised manuscript (Figure 6a, Figure 7, Figure 8a, Figure 9).

Reviewer 2 Report

The authors have prepared a manuscript about the incorporation of MXene into composite membranes for nanofiltration. This work complements a previous one published by the authors (Reference 14 in the manuscript) where evaluated the performance of MXene nanocomposite nanofiltration membranes for desalination. The main difference between both works is the synthesis procedure of the membrane. While in this work the authors propose the addition of MXene with the organic phase of interfacial polymerization, the previous work was focused on the addition of MXene in the aqueous phase. I would like to have further information about the main novelties provided by the new approach and I would like to get more direct comparison of the obtained results in both cases.

Other comments:

Page 3/26 (third paragraph)

The authors comment that “the membranes are limited by the “trade off” phenomenon and the serious operating environments”. Could the authors explain more clearly this “trade off” phenomenon?

Page 7/26 (first paragraph)

The authors describe the amounts of MXene in %, but the values are too low. I consider ppm can be a more adequate unit.

Page 9/26 (first paragraph)

In the desalination test of high-concentration brine, the test pressure was adjusted to 6 bars, while in the rest of experiments 4 bars were applied. I would like to know the reason why 6 bars were not applied to all the tests. This way, the direct comparison can be done more easily.

Page 10/26 (first paragraph)

The definition of J0 is not clear. Is it the initial permeation flux? is it the average of the previous three times?

Figure 7

The trend is very irregular. Have the authors tested that the differences are statically significant?

Figure 8

The authors only present the results of the M0 and M3 membranes. What about the rest of membranes?

Figure 9

The evolution of flux throughout time should be also provided.

Author Response

Reviewer 2’s Comments

The authors have prepared a manuscript about the incorporation of MXene into composite membranes for nanofiltration. This work complements a previous one published by the authors (Reference 14 in the manuscript) where evaluated the performance of MXene nanocomposite nanofiltration membranes for desalination. The main difference between both works is the synthesis procedure of the membrane. While in this work the authors propose the addition of MXene with the organic phase of interfacial polymerization, the previous work was focused on the addition of MXene in the aqueous phase. I would like to have further information about the main novelties provided by the new approach and I would like to get more direct comparison of the obtained results in both cases.

Response: Thanks for your professional comment. This new method is to embed MXene nanosheets in the organic phase. The main purpose is to hope that a small amount of MXene nanosheets will be covered by the polyamide layer, so that more MXene nanosheets will appear on the surface of the membrane, and the negative charge of the membrane will be directly enhanced. Because of this, the Na2SO4 rejection of the membrane prepared by embedding MXene in the organic phase is higher than that in the aqueous phase. Under the new research, the membrane also has better separation performance for high-concentration brine, monovalent and divalent mixed salts. However, the permeation flux of the membrane embedded with MXene in the aqueous phase is higher than that in the oil phase, and it has better stability, and the test can last for 58 days. We speculate that MXene is beneficial to enhance the stability of the membrane in the aqueous phase, and the existence of nano-transport channels is beneficial to increase the flux of the membrane. The membrane prepared by embedding MXene in the organic phase has a stronger Donnan effect on the membrane surface and thus has a higher rejection. This conjecture needs to be verified by our follow-up research.

Other comments:

  1. Page 3/26 (third paragraph)

The authors comment that “the membranes are limited by the “trade off” phenomenon and the serious operating environments”. Could the authors explain more clearly this “trade off” phenomenon?

Response: The “trade off” phenomenon refers to the balance between permeability and selectivity. When the membrane flux is increased, a part of the salt rejection rate will always be lost.

References: How to coordinate the trade-off between water permeability and salt rejection in nanofiltration? https://doi.org/10.1039/D0TA02510K

  1. Page 7/26 (first paragraph)

The authors describe the amounts of MXene in %, but the values are too low. I consider ppm can be a more adequate unit.

Response: We had made a correction following your suggestion. The revised version is visible in the revised manuscript.

  1. Page 9/26 (first paragraph)

In the desalination test of high-concentration brine, the test pressure was adjusted to 6 bars, while in the rest of experiments 4 bars were applied. I would like to know the reason why 6 bars were not applied to all the tests. This way, the direct comparison can be done more easily.

Response: Thank you for your rigorous comment. For our test of high-concentration brine, we first used 4 bars pressure, but later found that at 10000ppm, due to the stronger concentration polarization effect of high-concentration brine, 4 bars pressure was not conducive to collecting permeate, so we increased the pressure to 6 bars. In order to allow other readers to dispel doubts, we have made supplementary explanations in the paper before.

  1. Page 10/26 (first paragraph)

The definition of J0 is not clear. Is it the initial permeation flux? is it the average of the previous three times?

Response: J0 is the average of the previous three times. For long-term testing, we believed that the initial single flux could not accurately represent the actual initial flux due to test errors, etc., so the first three fluxes at the beginning were taken as the average value, which was also explained in the paper.

  1. Figure 7

The trend is very irregular. Have the authors tested that the differences are statically significant?

Response: The trend of these four kinds of salts is roughly that as the amount of MXene added increased, the salt rejection of the membrane gradually increased, which was mainly due to the further enhancement of the negative charge of the membrane surface. At a concentration of 180 ppm, the salt rejection of MXene membrane reached its maximum. When the concentration of MXene exceeded 180 ppm, the salt rejection of the membrane decreased slightly, mainly due to the agglomeration caused by the increase of MXene addition, which caused defects on the membrane surface. Since we found a suitable amount of MXene to add, we did not do a significant difference test.

  1. Figure 8

The authors only present the results of the M0 and M3 membranes. What about the rest of membranes?

Response: We appreciate for Reviewer’s comment. Based on the previous desalination test results, we determined the MXene membrane with the highest rejection. Following the current commonly used research methods, only the membrane with the best rejection was selected for MWCO, high-concentration brine desalination, and monovalent and divalent mixed salt system tests.

  1. Figure 9

The evolution of flux throughout time should be also provided.

Response: We had made a correction following your suggestion. The revised version is visible in the revised manuscript (Figure 9).

Reviewer 3 Report

This article reported the synthesis of MXene nanocomposite nanofiltration membrane and its salt removal performance. The work is interesting, and the manuscript is well written. A minor revision is needed before considering this paper for publication.

Below are specific comments. 

  1. The similarity is 26%. It should be reduced to the level of scientific publication.
  2. The abstract needs improvement. The details on page 1 should be condensed to a couple of sentences and the major potion should be the text starting on page 2.
  3. In Fig. 3a, the relevant peaks should be identified. The same should be done in Fig. 6a.
  4. The resolution of the figures should be enhanced. Some figures appeared as blurred images.
  5. It is suggested to include a schematic of the mechanism of flirtation and removal through the membrane.
  6. The following relevant references are suggested to be included.
  • 1039/C9EN01478K
  • https://doi.org/10.1016/j.cej.2020.124340
  • https://doi.org/10.1016/j.desal.2021.115448
  • https://doi.org/10.1007/s40820-020-0411-9

Author Response

Reviewer 3’s Comments

This article reported the synthesis of MXene nanocomposite nanofiltration membrane and its salt removal performance. The work is interesting, and the manuscript is well written. A minor revision is needed before considering this paper for publication.

Below are specific comments.

  1. The similarity is 26%. It should be reduced to the level of scientific publication.

Response: Thank you for your rigorous comment. The similarity of revised version is 14%, and the relatively high part of the repetition is focused on materials and methods, mainly because the materials, film formation method and test method were similar to the previous research.

  1. The abstract needs improvement. The details on page 1 should be condensed to a couple of sentences and the major potion should be the text starting on page 2.

Response: We follow your comments and simplify the abstract. The revised version is visible in the revised manuscript. Compared with the previous abstract, the new abstract has reduced 104 words.

  1. In Fig. 3a, the relevant peaks should be identified. The same should be done in Fig. 6a.

Response: We appreciate for Reviewer’s comment. We had made a correction following your comment. The revised version is visible in the revised manuscript (Fig. 3a, Fig. 6a).

  1. The resolution of the figures should be enhanced. Some figures appeared as blurred images.

Response: Thanks for your kind suggestion. We had reformed some figures following your suggestion. The revised version is visible in the revised manuscript (Figure 1e, Figure 2a, Figure 2c, Figure 6d, Figure 6c, Figure 8b, Figure 8d).

5.It is suggested to include a schematic of the mechanism of flirtation and removal through the membrane.

Response: We follow your comments. In revised version, we have a brief discussion on its mechanism in each separation performance,

  1. It is suggested to include a schematic of the mechanism of flirtation and removal through the membrane.

The following relevant references are suggested to be included.

1039/C9EN01478K

https://doi.org/10.1016/j.cej.2020.124340

https://doi.org/10.1016/j.desal.2021.115448

https://doi.org/10.1007/s40820-020-0411-9

Response: We appreciate for Reviewer’s comment. These reviews summarize very complete studies on mechanism of flirtation and removal through the MXene membrane, not only good for this research but also for our future research.

Revision made: Due to its strong negative charge, excellent hydrophilicity and adjustable properties, it has been widely used in the membrane modification to improve desalination performance [9-13].

This might be due to the increase in the cross-linking degree and negative charge in the membrane surface with negative MXene, which increased the rejection of divalent anions[25].

The high rejection of divalent ions and the low rejection of monovalent ions might be due to the adsorption and aggregation of monovalent ions in the ion selective channels of MXene nanosheets, which reduced the size of ion channels and made it difficult for divalent ions to pass through [29].

References added in revised version:

[13] I. Ihsanullah, MXenes (two-dimensional metal carbides) as emerging nanomaterials for water purification: Progress, challenges and prospects, Chemical Engineering Journal, 388 (2020) 124340.https://doi.org/10.1016/j.cej.2020.124340

[25] O. Kwon, Y. Choi, J. Kang, J.H. Kim, E. Choi, Y.C. Woo, D.W. Kim, A comprehensive review of MXene-based water-treatment membranes and technologies: Recent progress and perspectives, Desalination, 522 (2022) 115448.https://doi.org/10.1016/j.desal.2021.115448

[29] I. Ihsanullah, Potential of MXenes in Water Desalination: Current Status and Perspectives, Nano-Micro Letters, 12 (2020) 72.https://doi.org/10.1007/s40820-020-0411-9

Round 2

Reviewer 2 Report

The authors have answered adequately to the comments done in the initial review round. However, these answers have not been included in the manuscript. I consider that further information about these justifications should be included in the main text of the manuscript.

The language of the revised texts must be improved.

The caption of Figure 9 must be replaced to include a mention to the evolution of the permeate flux.

Author Response

Reviewer’s comments:

The authors have answered adequately to the comments done in the initial review round. However, these answers have not been included in the manuscript. I consider that further information about these justifications should be included in the main text of the manuscript.

1.The language of the revised texts must be improved.

Response: We followed your kind comments and revise some discussions as possible as we can. The revised version is visible in the revised manuscript.

2.The caption of Figure 9 must be replaced to include a mention to the evolution of the permeate flux.

Response: Thanks for Reviewer’s comment. We had made a correction following your comment. The revised version is visible in the revised manuscript.